# Sex Differences in Incidence, Clinical Characteristics and Outcomes in Children and Young Adults Hospitalized for Clinically Suspected Myocarditis in the Last Ten Years—Data from the MYO-PL Nationwide Database

**DOI:** 10.3390/jcm10235502

**Published:** 2021-11-24

**Authors:** Krzysztof Ozierański, Agata Tymińska, Aleksandra Skwarek, Marcin Kruk, Beata Koń, Jarosław Biliński, Grzegorz Opolski, Marcin Grabowski

**Affiliations:** 1First Department of Cardiology, Medical University of Warsaw, 02-097 Warsaw, Poland; krzysztof.ozieranski@wum.edu.pl (K.O.); S073784@student.wum.edu.pl (A.S.); grzegorz.opolski@gmail.com (G.O.); grabowski.marcin@me.com (M.G.); 2National Health Fund, 02-528 Warsaw, Poland; Marcin.Kruk@nfz.gov.pl (M.K.); Beata.Kon@nfz.gov.pl (B.K.); 3Departament of Haematology, Transplantation and Internal Medicine, Medical University of Warsaw, 02-097 Warsaw, Poland; jaroslaw.bilinski@gmail.com

**Keywords:** cardiomyopathy, endomyocardial biopsy, epidemiology, heart failure, arrhythmias, inflammation

## Abstract

There is a widespread lack of systematic knowledge about myocarditis in children and young adults in European populations. The MYO-PL nationwide study aimed to evaluate sex differences in the incidence, clinical characteristics, management and outcomes of all young patients with a clinical diagnosis of myocarditis, hospitalized in the last ten years. The study involved data (from the only public healthcare insurer in Poland) of all (*n* = 3659) patients aged 0–20 years hospitalized for myocarditis in the years 2011–2019. We assessed clinical characteristics, management and five-year outcomes. Males comprised 75.4% of the study population. The standardized incidence rate of myocarditis increased over the last ten years and was, on average, 7.8 and 2.5 (in males and females, respectively). It was the highest (19.5) in males aged 16–20 years. The highest rates of hospital admissions occurred from late autumn to early spring. Most myocarditis-directed diagnostic procedures, including laboratory tests, echocardiography, coronary angiography, cardiac magnetic resonance and endomyocardial biopsy, were performed in a low number of patients, particularly in females. Most patients required rehospitalization for cardiovascular reasons. The results of this large epidemiological study showed an increasing incidence of myocarditis hospitalizations in young patients over last ten years and that it was sex-, age- and season-dependent. Survival in young patients with myocarditis was age- and sex-related and usually it was worse than in the national population. The general management of myocarditis requires significant improvement.

## 1. Introduction

Myocarditis in children has become an increasingly significant condition over the years. Myocarditis can take various forms, from a mild subclinical course to fulminant acute myocarditis, entailing acute heart failure and being the cause of sudden cardiac death in 4–6% of cases [1,2,3]. In addition, a number of major complications have been observed, such as dilated cardiomyopathy or clinically significant arrhythmias [4]. The Global Burden of Disease 2016 and 2019 Study (GBD2016 and GBD2019) show a higher-than-expected increase in all-age deaths due to myocarditis over the last decades [5,6]. However, there is still a substantial gap in the current knowledge of the epidemiology, diagnosis, treatment and true course of the disease not only in adults but also in the pediatric/young adult population.

The incidence, management and outcomes of myocarditis in clinical practice in hospitalized patients aged 0–20 years are largely unknown in European populations. There are several nationwide studies investigating myocarditis in pediatric populations; however, there is only one such study performed on a European population [7]. Based on the examples of American or Asian studies, which provided valuable information and conclusions, we believe that big, nationwide databases are of especially high scientific value, as they report real-life, unselected data on the clinical characteristics of patients and on current diagnostic standards.

Several large-scale studies on myocarditis/cardiomyopathy in children have been conducted so far. However, they provide conflicting results due to non-homogenous diagnostic criteria and possible ethnic differences [1,2,3,4,7,8]. The available data seem to show an increase in the incidence rate of myocarditis in pediatric patients in the last decade [2]. However, most data were collected prior to 2016 and require updating. What is more, there is little information regarding the sex- and age-related short- and long-term outcomes, without explanation of the possible rationale behind the differences [2,3]. In addition, only one of the abovementioned studies examined the European population, and thus, there is a particular need for studies from this region [7].

Considering the abovementioned lack of knowledge and the rising significance of myocarditis, particularly in the pediatric population, obtaining more population data, particularly from European countries, is highly warranted. In this analysis of a nationwide MYO-PL (the occurrence, trends, management and outcomes of patients with myocarditis in Poland) database containing information about all patients with myocarditis, we aimed to evaluate sex-related differences in the incidence, clinical characteristics, management and long-term outcomes of all real-life, unselected patients aged 0–20 years with a clinical diagnosis of myocarditis hospitalized in Poland in the last ten years.

## 2. Materials and Methods

The MYO-PL nationwide study gathered data from the National Health Fund (NHF) —the only public healthcare insurer in Poland. The NHF reimburses healthcare services (both public and private) and prescription of medication with public funds. Public health insurance is obligatory for almost all Polish residents. More research results have already been published based on the NHF data regarding the incidence and management of certain diseases [9,10].

Based on the NHF data, we derived hospitalizations due to myocarditis reported in the years 2011–2019 with the following ICD-10 (the International Classification of Diseases and Related Health Problems, 10th Revision) codes: I40, I40.0, I40.1, I40.8, I40.9, I41, I41.0, I41.1, I41.2, I41.8, I51.4 and B33.2 [5]. The diagnostic criteria of myocarditis (based on international ICD-10 codes) were clinician-dependent and reflected routine clinical practice. No other inclusion–exclusion criteria were applied.

The dataset was restricted to the first hospitalization for myocarditis (newly diagnosed patients with a principal (first) diagnosis of myocarditis for whom no information about myocarditis was reported in the 400 days preceding the hospitalization). To establish the baseline clinical characteristics, we analyzed the data for each patient 400 days back dating from the initial diagnosis of myocarditis. In addition, in-hospital and long-term outcomes were analyzed, including all-cause mortality as well as the occurrence of selected diseases (defined as receiving services with specific ICD-10 codes reported) and selected procedures, defined with particular codes following the International Classification of Diseases, 9th Revision, Clinical Modification (ICD-9-CM) (all ICD codes used in this analysis are presented in a Appendix A) [6]. The follow-up was assessed for the in-hospital period, 30-days, one-year, three-years, and five-years after discharge. Thus, only patients with a diagnosis of myocarditis between January 2011 and December 2019 were ultimately included in the analysis.

To show age-related differences in patients hospitalized due to myocarditis, baseline characteristics and long-term outcomes were assessed for the following age groups: 0–5, 6–10, 11–15 as well as 16–20 years. Patients aged 0–20 years were included in the study to show age-related differences in children and young adults. The incidence of myocarditis was calculated by the number of residents (per 100,000 people) in Poland in a given age group in the years 2011–2019. Age-standardized hospitalization rates for myocarditis in male and female patients were calculated in relation to the age structure of the Polish population (aged 0–20) from 2011.

No ethics approval was required for this study, as it involved the analysis of administrative data. The study complies with the Declaration of Helsinki.

In the study, data from the Central Statistical Office of Poland was used to relate the obtained results to the Polish population and to build life tables for survival analysis.

### Statistical Analysis

The results were presented as means (and standard deviations) or median (and quartiles) for continuous variables. Categorical variables were presented as percentages. The Shapiro–Wilk test was performed to verify the normality of the data distribution. Median values and Mood’s median test or mean values and Welsch’s *t*-test were used in case of normal and non-normal data distribution, respectively. Associations between the study parameters were analyzed using a Pearson chi-square test and a *t*-test. The observed survival was analyzed using the Kaplan–Meier estimates. The relative survival (with 95% CIs) was calculated using the Hakulinen method employing single age-, year-, and sex-specific life tables for the general Polish population. Logistic regression models were created to report age-adjusted sex differences. A *p*-value less than 0.05 was considered significant. All tests were 2-tailed. Statistical analysis was carried out using R software, version 3.6.1 (Columbus, OH, US)

## 3. Results

### 3.1. Incidence of Hospitalizations for Myocarditis and Clinical Characteristics

Medical records of 3659 hospitalized children with a clinical diagnosis of myocarditis were collected between 2011 and 2019. The median age of the total cohort was 17 years (interquartile range 8–19), and was 10 (1–16) years in males and 17 (13–19) years among females (*p* < 0.001). The majority of the patients were male (75.4%, *n* = 2759).

The standardized incidence rate of myocarditis was, on average, 7.8 and 2.5 (in males and females, respectively) and was the highest in males aged 16–20 years. The incidence of myocarditis hospitalizations showed a bimodal distribution, with two peaks: lower in children, aged 0–5 years and higher in young adults, aged 16–20 years (Table 1).

An increase in age-standardized incidence rates of myocarditis hospitalizations over the study period was observed, but it was driven by a substantial increase in the myocarditis incidence rate in males (Figure 1). Conversely, in females, a slight decrease in the incidence rate was seen over time.

The incidence rates of myocarditis were higher in males in nearly all age groups and all study years (Table 1). However, the sex difference in the occurrence of myocarditis was clearly age-related. The difference in favour of higher incidence rates of myocarditis in males increased with age and was the highest in patients aged 16–20 years.

We observed a pattern of seasonal changes in the frequency of hospitalizations for myocarditis (Figure 2). The highest rates of hospital admissions occurred from late autumn to early spring (November to April), while the lowest rates were observed in mid-summer (July to August).

Serious cardiac arrhythmias (ventricular tachycardia, ventricular fibrillation and atrial fibrillation) before admission were infrequent both in male and female patients. In females, compared to males, higher rates of tachycardia/palpitations (3.3% vs. 1.2%, respectively) and paroxysmal tachycardia (1.3% and 0.3%, respectively) were observed.

Females were more likely to have had a history of infectious diseases within the last 6 months prior to admission, especially otorhinolaryngologic and/or ophthalmic infections (46.4% vs. 41.0% in males; *p* < 0.01) as well as digestive infections (7.6% vs. 4.8% in males; *p* = 0.01).

### 3.2. In-Hospital Diagnostic Procedures and Management

Most patients were hospitalized in general wards (internal ward or pediatric department) (49.7%) and in cardiology units (40.7%), but significant sex differences were evident. Males were more frequently admitted to cardiology units (46.4%) in comparison to females (23.3%) who were mostly hospitalized in general wards (59.8% compared to 46.4% in males). Females were more frequently hospitalized in intensive care units (3.8%) than males (1.1%) (Table 2).

Most of the myocarditis-directed diagnostic procedures, including laboratory tests, echocardiography, invasive/computed tomography coronary angiography, cardiac magnetic resonance (CMR) or endomyocardial biopsy (EMB), were performed in a low number of patients, especially in females (Table 2). Laboratory studies such as C-reactive protein (43.2% vs. 33.4%; *p* < 0.01), troponins (40% vs. 27.2%; *p* < 0.01) and brain natriuretic peptides (14.4% vs. 8.8%; *p* < 0.01) were more frequently performed in males as compared to females. Males were also more likely to undergo echocardiography (88.3% vs. 83.7%; *p* < 0.01) and/or coronary angiography (9.4% vs. 1.1%; *p* < 0.01) than females. There were no significant sex-related differences in the frequency of CMR and EMB performance, but overall, these procedures were performed in a minority of patients (15.4% and 0.3%, respectively). Consequently, it can be concluded that the etiology of myocarditis remained uncertain in the majority of cases.

The main clinical characteristics and management approaches for patients with myocarditis are presented in Table 2.

### 3.3. Short- and Long-Term Outcomes

During the five-year observation period, most patients required rehospitalization. The mean number of hospitalizations in a five-year follow-up were 2.8 and 2.0 (*p* = 0.001) for females and males, respectively. Female patients were more frequently hospitalized due to cardiac arrhythmias and cardiomyopathy/heart failure, while male patients were more likely to be rehospitalized for myocarditis (however, there were no statistically significant sex-related differences in the reasons for hospitalization) (Table 3).

There were no differences between male and female patients in the short-term (in-hospital and 30-day) observed mortality regardless of the age group. However, in the long-term follow-up (after five-years), a higher mortality in females compared to males was seen in the group aged 0–5 years (6.4% vs. 1.3% (*p* = 0.01), respectively) (Table 4 and Figure 3). There were no differences in mortality between male and female patients in other age groups. The relative five-year survival rate ranged from 0.96 to 0.99 in males and from 0.95 to 0.99 in females. The relative survival rates in females aged 0–5 and 6–10 years were worse, while in females aged 11–15 and 16–20 years, they were equal/not worse compared with the general population. Relative survival rates in males aged 0–5 years were equal/not worse compared with the general population. In other age groups, the relative survival rates in males were mostly worse than in the general population. Relative survival rates in relation to sex and age are shown in Table 5.

## 4. Discussion

### 4.1. The Current State of Knowledge and Novelty of the Study

Data on myocarditis in the pediatric population in Europe is highly insufficient. This is especially true for the outcomes of myocarditis and the prognostic factors, which are largely unknown. Most data come from studies on non-European populations, with only one European study based on a nationwide database, performed in Finland few years ago. In Poland, a recent national database provided valuable conclusions about the incidence, presentation and outcomes of myocarditis in the general population [4]. However, these data do not allow for extrapolation for pediatric populations.

This leaves a significant gap in the current general knowledge. MYO-PL is a nationwide study that includes the unique data of all patients aged 0–20 years hospitalized with a clinician-based diagnosis of myocarditis in the last ten years in a large (approx. 38 million people) European country. This study was conducted on what is, to date, the largest European pediatric population. It also presents mostly unknown data regarding sex-related differences and outcomes in short- and long-term follow-up. Moreover, the reduction in the age span of the study groups to 5 years only allows for a better insight into the characteristics of the patients with regard to age groups and displays the age differences more accurately. This database provides up-to-date and comprehensive information concerning the real-life epidemiology, management and long-term course of the disease in daily clinical practice.

What constitutes the novelty of this study is the detailed presentation of the sex-related differences in the incidence, clinical characteristics and outcomes, both short- and long-term. We proved that the overall incidence was higher in males, with the sex-related difference being most accentuated in the group aged 16–20 years.

### 4.2. Incidence and Patient Characteristics

Previously published research concerning the epidemiology of myocarditis in children in the western population reported the incidence rates spanning a broad spectrum of values. The incidence rates amounted to: 1.95/100,000 person-years in Finland [7], 1.4–2.1/100,000 in Korea [8], 0.8/100,000 in United States (US) [11] and 0.26/100,000 in Japan [4]. The age spans of the investigated groups were 0–15 years, 0–19 years, 0–18 years and 1 month–17 years, respectively. The data were derived from registries [7,8,11] or questionnaires [4] reporting patients hospitalized in the years 2004–2014, 2007–2016, 2007–2016 and 1997–2002, respectively. These studies included patients with a clinical (not EMB-based) diagnosis of myocarditis based on ICD-10 codes (Korean, Finnish and US studies) [7,8] and diagnostic criteria established by the Japanese Circulation Society [4], with no specific exclusion criteria. Importantly, a bimodal age distribution was shown in the studies from Korea and the US by Vasudeva et al. [11], with two peaks occurring, respectively, in infancy and in young adulthood, both in males and females, which is also consistent with the results of our study.

Retrospective data from the registry by Vasudeva et al. suggest a gradual upward tendency of myocarditis hospitalization occurrence with 2898 admissions in 2007 and 3625 in 2014 (27,129 hospitalizations overall during the study period) [12]. A similar increase was observed in a study on patients aged 0–19 years, which demonstrated that the incidence rate increased significantly from 1.4 to 2.1 per 100,000 patients over a nine-year period in the Korean population [8]. Notably, at the study endpoint (2016), the incidence showed a significant sex-related difference (2.5 per 100,000 boys and 1.7 per 100,000 girls, respectively). In our study, we also proved that the increase over time was driven by the male population, mainly at age 16–20, with a slight decrease in the female population. This contrasts with the previous findings by Vasudeva et al. [11], according to which there was a homogenous increase in the incidence in all age and sex groups.

Our study showed progressive male predominance, increasing with age. This is consistent with the previously published studies. In the Korean Study, this difference was demonstrated in groups of patients older than 13 years, whereas in the Finnish Study, as well as in ours, it was detected already in the groups of patients aged 6–11 and 6–10 years, respectively, with a further increase in older patients [7,8]. As the study cohort in the Finnish Study was aged 0–15 years, data limitations need to be considered regarding the young adult population. This problem was addressed in our study by enrolling into the study a group aged 16–20 years, where the sex-related differences turned out to be most pronounced. With regard to the sex-related differences, females were reported to be, on average, younger than males at the disease onset (mean age in females was 10.6 [7] and 6.5 [8], and in males was 12.8 [7] and 9.8 [8]), which was also corroborated in our study (median age was 10 years in females and 17 years in males).

This specific age distribution and predominance of myocarditis occurrence in males is still an unexplained phenomenon that requires further research in order to determine whether it is attributable to a factor that cannot be affected, such as sex-related hormonal balance, or whether it is caused by a possibly modifiable environmental factor.

There are several theories as to what contributes to this distinctive sex-related difference. Some of them suggest a substantial role of sex hormones influencing several components of the immune system. Epidemiological data suggest that estrogen might have a protective character, while testosterone may promote the occurrence of myocarditis. It has been experimentally proven that estrogen and testosterone can modify the reactions of various immune cell populations [13,14]. The influence of genetic factors also needs to be considered. Some authors also attribute these sex-related differences to exercising and physical activity reaching higher intensity among males (particularly relevant in young adulthood) [7].

On the other hand, the infectious factor (mainly viruses) is the most often raised reason for the occurrence of myocarditis [3,15]. In our study, the highest incidence of myocarditis in males was observed from November to April when the impact of infectious etiology of myocarditis might be especially expected. Interestingly, in our study, the frequency of a recent history of an infectious disease was higher in females, while the incidence rate of myocarditis was higher in males, suggesting that possible non-infectious factors may be crucial in the development of a myocardial inflammation. The pathophysiological mechanisms responsible for this specific sex- and age-related distribution remain unknown and require further research.

Furthermore, one may hypothesize that different factors play major roles in US and European populations, as the increase in myocarditis hospitalizations over the years has shown a different pattern. Whereas in the US, a homogenous increase in all age and sex groups was observed, in Europe, the group of young, male adults was especially affected, suggesting the greater role of environmental factors in US, with a more complex etiology (including possibly hormonal and/or immunological changes or influence of physical activity, that are most pronounced during adolescence) in Europe.

The observed higher rate of arrhythmias in females with myocarditis is a so-far uninvestigated phenomenon. Several studies on non-myocarditis patients reported a higher propensity towards arrhythmias (especially supraventricular arrhythmias) in adult females compared to males and linked it with longer rate-corrected QT-intervals in females [16,17,18]. This has been attributed in many studies to the influence of sex hormones, which does not, however, explain the propensity for arrhythmia in pediatric populations, as hormone differences are less pronounced until puberty [19]. The lower density of ionic currents responsible for early repolarization (mainly K+ channels) in female hearts has been suggested as an underlying cause of the higher observed rate of arrhythmias in females. However, this conclusion is based solely on experiments on rabbit hearts and requires validation in further studies [13,14,20]. It is likely that heart inflammation may aggravate these proarrhythmic mechanisms in females.

### 4.3. In-Hospital Diagnostic Procedures and Management

We also demonstrated that the application of standard diagnostic procedures required for the confirmation of the diagnosis was generally very low. Interestingly, females were more likely to be admitted to general wards, and not to receive the routine cardiological diagnostic tests (lab tests, echocardiography and coronary angiography) in comparison to males. It should be noted that this tendency of limited testing was more visible in females, even though they more often presented with palpitations and/or tachycardia. In general, the use of golden standard non-invasive (CMR) and invasive (EMB) procedures was marginal. Particularly, the clinical utility of EMB in the diagnosis of myocarditis should be highlighted. EMB provides the confirmation of the diagnosis as well as details on the etiology and type of inflammatory cell infiltration. EMB is associated with a relatively low risk of serious complications (1.9%) as shown in a large retrospective study on a pediatric population [21]. However, the performance of EMB in a center with expertise in this field is strongly recommended [22]. The results of our study confirm that there is a pressing need for the optimization of the diagnostic standards, both in male and female patients, at the level of the healthcare system. Females were more frequently hospitalized in wards other than cardiology wards (i.e., internal ward or pediatric department) and tended to have a worse in-hospital prognosis in most age groups compared to males.

### 4.4. Outcomes

Short- and long-term outcomes in children and young adults are mostly unknown. The GBD2016 and GBD2019 reports showed a much higher number of life-years lost than expected in the populations of eastern and central Europe [5,6]. In the Finnish study, the overall in-hospital mortality was 1.4%, but the small group of patients precluded statistical analyses of age- and sex-differences [7]. In our study, the in-hospital mortality was lower (<1%) in older (aged over 10 years) than in younger patients (1–2.5%). There were no differences between male and female patients.

In the available literature, there are also no data on long-term outcomes regarding sex-related differences. The Japanese study showed an overall 22.5% mortality (38 out of 169 patients with a one-year follow-up) while the Korean study demonstrated that as many as 30.5% of patients died or had heart transplantation (113 out of 371 patients during the peri-hospital period); however, the latter study included only patients with acute fulminant myocarditis [8]. The strength of our study is in the comprehensive data it provides on five-year outcomes (with data on relative survival based on the total Polish population), rendering it perfect for epidemiological considerations and showing large-scale trends in the outcomes of myocarditis in children and young adults. Compared to the general Polish population, the survival rates in our study tended to be worse in the youngest groups of females (aged 0–5 and 6–10 years) and in the older groups of males. The observed long-term mortality was the highest in patients aged 0–5 years, and was higher in females than in males. There were no differences in other age groups.

The number of rehospitalizations tended to be higher in females, which is a variable that had not been investigated in previous studies. However, there were no significant differences in the reason for hospitalization between male and female patients.

This may be due to the previously described sex differences, encompassing hormonal and immunological differences, which can not only affect the susceptibility to infection and development of the disease, but also the severity and the risk of rehospitalization. Furthermore, the fact that girls are more often hospitalized within general wards may have a negative influence on the quality of care and, consequently, lead to more frequent rehospitalizations.

The importance of this study for public health research is not to be underrated. We demonstrated sex-related differences in the diagnostics and treatment, as well as the outcomes of myocarditis, which provides a unique opportunity for a thoughtful analysis.

With this study, we also provide “hypothesis-generating” material for further research on age- and sex-differences in myocarditis. The main questions remaining to be answered are: ‘What makes males more susceptible to myocarditis?’ and ‘Why is the occurrence of myocarditis age dependent?’. The answers to these questions may also provide more data on true etiological factors (infectious vs. non-infectious and immune-mediated) of myocarditis.

### 4.5. Diagnosis of Myocarditis in the COVID-19 Era

Results of our study are particularly important and should be discussed in the light of a recent Severe Acute Respiratory Syndrome Coronavirus-2 pandemic. At the moment, the consensus has been reached that it is not a cardiotropic virus and acute myocarditis represented by the viral presence and infiltrates in the myocardium with myocardial injury is rarely observed in the course of Coronavirus Disease 2019 (COVID-19) [15,23]. In individuals with so far uninvestigated predispositions the virus can be a trigger for an immune reaction, leading to immune-mediated myocarditis. 

In order to avoid misdiagnosis of myocarditis in patients with COVID-19, we recommend implementing the structured diagnostic algorithm, included in the guidelines created by European Society of Cardiology [24,25]. It should be highlighted that nasal/throat swab tests for COVID-19 in the presence of clinically suspected myocarditis does not allow for establishing the etiology. COVID-19 pandemic did not change the diagnostic path and still EMB/autopsy is a gold standard of confirming the diagnosis.

### 4.6. Limitations

It is a limitation of the study that the diagnosis of myocarditis was made on the basis of the clinical presentation and auxiliary diagnostic measures, as EMB was very rarely performed, and thus, no histological confirmation was possible. The criteria for the diagnosis of myocarditis (as well as access to advanced diagnostic procedures, i.e., CMR and EMB) have changed over time, making the diagnosis of myocarditis usually a diagnosis of exclusion. Myocarditis should be confirmed by EMB; however, worldwide EMB is performed only in selected centers and still not in the majority of patients. However, the incidence rates of myocarditis presented in our study were in-line with previous similar studies. In contrast to clinical trials, the character of the study is also associated with certain data inaccuracy, dependent on physicians reporting the diseases and procedures using ICD codes. Due to the fact that data were extracted from the national health care payer, we were not able to neither verify each diagnosis nor provide exact results of diagnostic tests, which is a limitation inherent to this type of data collection. Nevertheless, while analyzing myocarditis, one needs to base the observations on real-life data of unselected patients, as standards from clinical trials rarely correspond to routine clinical practice.

## 5. Conclusions

The results of this large epidemiological study showed an increasing incidence of myocarditis in children and young adults over the last ten years, and that it was sex-, age- and season-dependent. It also made it clear that the management of myocarditis requires significant improvement at the national level. The overall survival in young patients with myocarditis was age- as well as sex-related, and was usually worse than in the general population. The observed mortality was the worst in the youngest females.

## Figures and Tables

**Figure 1 jcm-10-05502-f001:**
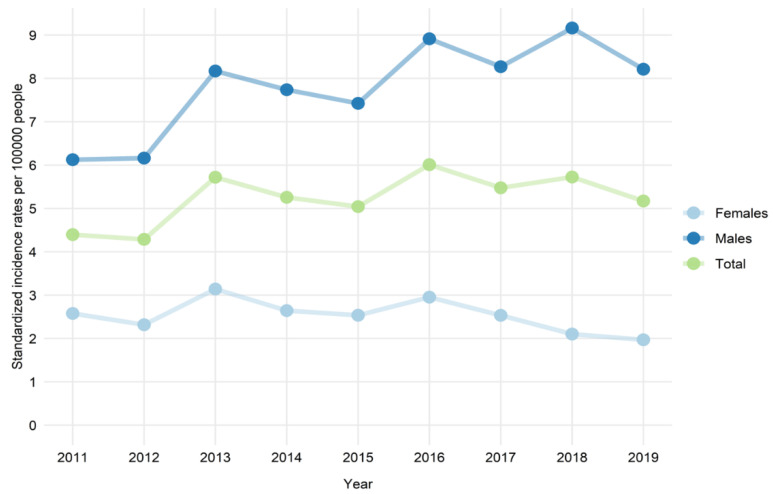
Age-standardized hospitalization rates for myocarditis in male and female patients aged 20 years or younger by the number of residents in Poland (per 100,000 people) in the years 2011–2019.

**Figure 2 jcm-10-05502-f002:**
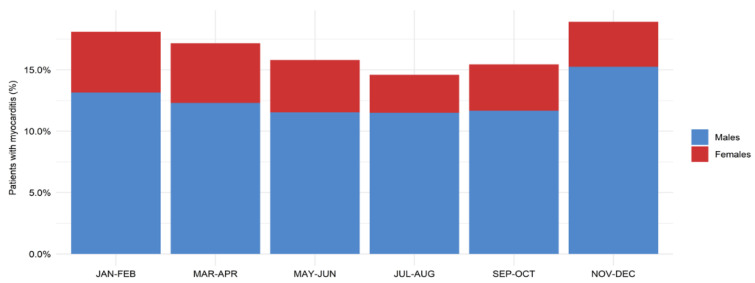
Seasonality of the incidence of male and female patients with myocarditis aged 20 years or younger. Chi-square test for the difference: *p*-value < 0.001.

**Figure 3 jcm-10-05502-f003:**
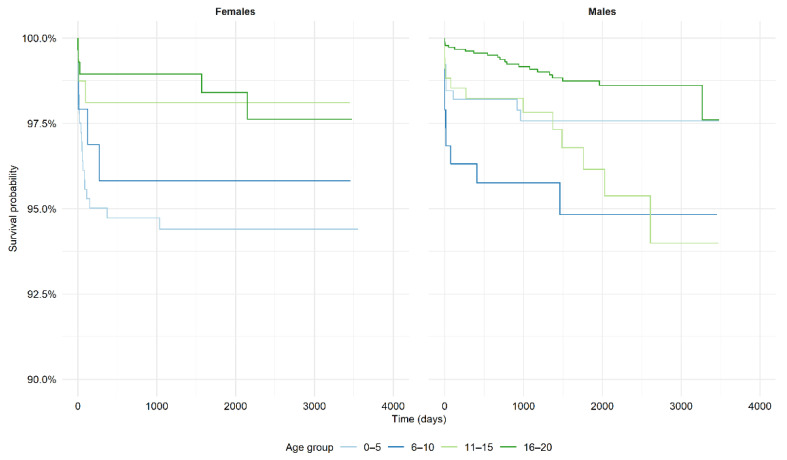
Kaplan–Meier curves for observed mortality in patients hospitalized with myocarditis.

**Table 1 jcm-10-05502-t001:** The incidence of male and female patients with myocarditis by the number of residents (per 100,000 people) in Poland in a given age group in the years 2011–2019.

Males												
Age Group (Years)	Average Incidence	Year									*p*-Value ^a^	*p*-Value ^b^
		2011	2012	2013	2014	2015	2016	2017	2018	2019		
0–5	3.56	4.28	4.43	4.09	4.52	3.46	2.73	2.90	3.40	2.20	0.003	<0.001
6–10	2.09	2.16	1.39	2.61	2.01	1.35	2.83	1.22	3.21	2.04	0.615	0.392
11–15	3.97	2.55	2.43	4.48	4.26	4.00	4.56	4.62	4.82	4.02	0.034	0.005
16–20	19.52	13.90	14.56	19.52	18.16	18.81	23.31	21.98	23.03	22.37	0.001	<0.001
**Females**												
**Age Group (Years)**	**Average Incidence**	**Year**									***p*-Value ^a^**	***p*-Value ^b^**
		**2011**	**2012**	**2013**	**2014**	**2015**	**2016**	**2017**	**2018**	**2019**		
0–5	3.49	4.26	3.67	5.00	3.73	3.12	3.25	3.60	2.70	2.06	0.009	0.001
6–10	1.11	0.80	0.56	1.54	1.16	0.81	1.39	1.09	1.59	1.02	0.249	0.221
11–15	1.95	1.03	1.28	2.19	2.58	2.05	2.29	2.49	1.98	1.69	0.241	0.148
16–20	3.09	3.46	3.09	3.23	2.70	3.62	4.36	2.57	1.98	2.81	0.324	0.216

^a^ Linear regression and ^b^ P-trend tests for the independence of the number of patients with myocarditis in a given age group by year.

**Table 2 jcm-10-05502-t002:** Clinical characteristics and diagnostic procedures performed in male and female patients aged 20 years or younger hospitalized for myocarditis.

Variable	Total *n* = 3659	Females *n* = 900 (24.6%)	Males *n* = 2759 (75.4%)	OR (95% CI) *	*p*-Value *
Demographics					
Median Age (IQR)	17 (8–19)	10 (1–16)	17 (13–19)	-	-
Management					
Hospital ward on admission, n (%) - Cardiology unit	1491 (40.7)	210 (23.3)	1281 (46.4)	0.57 (0.47–0.68)	<0.01
General ward (internal ward or pediatric department)	1817 (46.8)	538 (59.8)	1279 (46.4)	1.25 (1.06–1.48)	0.01
Intensive care unit	64 (1.6)	34 (3.8)	30 (1.1)	1.89 (1.12–3.18)	0.02
Intensive cardiac care unit	32 (0.9)	3 (0.3)	29 (1.1)	0.38 (0.11–1.31)	0.13
Other	475 (13.0)	136 (15.1)	339 (12.3)	1.17 (0.93–1.47)	0.18
Diagnostic procedures, n (%) - C-reactive protein **	1494 (40.8)	301 (33.4)	1193 (43.2)	0.79 (0.67–0.93)	<0.01
Troponins **	1349 (36.9)	245 (27.2)	1104 (40.0)	0.73 (0.61–0.87)	<0.01
Brain natriuretic peptides **	475 (13.0)	79 (8.8)	396 (14.4)	0.66 (0.51–0.87)	<0.01
Echocardiography ***	3188 (87.1)	753 (83.7)	2435 (88.3)	0.72 (0.57–0.90)	<0.01
Cardiac Magnetic Resonance ***	563 (15.4)	88 (9.8)	475 (17.2)	0.81 (0.63–1.05)	0.11
Endomyocardial biopsy ***	12 (0.3)	2 (0.2)	10 (0.4)	1.1 (0.23–5.26)	0.91
Endomyocardial biopsy or heart catheterization ***	40 (1.1)	13 (1.4)	27 (1.0)	1.43 (0.70–2.92)	0.32
Coronary angiography (invasive or computed tomography) ***	270 (7.4)	10 (1.1)	260 (9.4)	0.23 (0.12–0.43)	<0.01
Medical history					
Cardiac Arrhythmias, n (%) - Atrial Fibrillation	7 (0.2)	4 (0.4)	3 (0.1)	3.54 (0.71–17.62)	0.12
Tachycardia, palpitations	63 (1.7)	30 (3.3)	33 (1.2)	3.36 (1.97–5.74)	<0.01
Bradycardia	12 (0.3)	4 (0.4)	8 (0.3)	0.81 (0.23–2.79)	0.73
Paroxysmal tachycardia	19 (0.5)	12 (1.3)	7 (0.3)	5.67 (2.10–15.33)	<0.01
Ventricular tachycardia	2 (0.1)	1 (0.1)	1 (0.0)	7.33 (0.44–123.49)	0.17
Ventricular fibrillation	2 (0.1)	0 (0.0)	2 (0.1)	-	0.99
Heart failure, n (%)	19 (0.5)	6 (0.7)	13 (0.5)	1.08 (0.39–3.00)	0.89
Hypertension, n (%)	53 (1.4)	4 (0.4)	49 (1.8)	0.47 (0.17–1.33)	0.16
Diabetes, n (%)	17 (0.5)	4 (0.4)	13 (0.5)	1.70 (0.53–5.43)	0.37
Chronic kidney disease, n (%)	2 (0.1)	1 (0.1)	1 (0.0)	9.36 (0.57–153.2)	0.12
Asthma, n (%)	227 (6.2)	62 (6.9)	165 (6.0)	1.11 (0.81–1.54)	0.51
Autoimmune disease, n (%)	28 (0.8)	9 (1.0)	19 (0.7)	1.71 (0.73–4.00)	0.21
Psychiatric disease, n (%)	44 (1.2)	9 (1.0)	35 (1.3)	1.39 (0.65–2.99)	0.39
Infectious disease within last 6 months, n (%) - Otolaryngologic and eye	1548 (42.3)	418 (46.4)	1130 (41.0)	1.26 (1.07–1.48)	<0.01
Central nervous system	4 (0.1)	1 (0.1)	3 (0.1)	0.70 (0.07–7.44)	0.77
Respiratory	503 (13.7)	142 (15.8)	361 (13.1)	0.98 (0.78–1.22)	0.85
Digestive	200 (5.5)	68 (7.6)	132 (4.8)	1.58 (1.15–2.19)	0.01
Urogenital	30 (0.8)	11 (1.2)	19 (0.7)	1.12 (0.51–2.47)	0.77
Sepsis	14 (0.4)	4 (0.4)	10 (0.4)	0.81 (0.24–2.72)	0.73
Other	211 (5.8)	68 (7.6)	143 (5.2)	1.33 (0.97–1.83)	0.08
Any	1903 (52.0)	503 (55.9)	1400 (50.7)	1.25 (1.06–1.47)	0.01

* Comparison between sex groups, adjusted for age; ** within the index hospitalization; *** within last or proceeding 6 months from the diagnosis of myocarditis; CI—confidence interval; IQR—interquartile range; n—number; OR—odds ratio.

**Table 3 jcm-10-05502-t003:** Reasons for rehospitalization in the five-year follow-up in patients with myocarditis aged 20 years or younger according to sex.

Reason for Rehospitalization	Females (%)	Males (%)	OR *	95% CI	*p*-Value
Atrial fibrillation or atrial flutter	0 (0.0)	0 (0.0)	1.00	-	1.00
Ventricular fibrillation, ventricular flutter or cardiac arrest	1 (0.2)	4 (0.3)	0.59	0.06–5.77	0.65
Other cardiac arrhythmias or paroxysmal tachycardia	53 (10.8)	96 (7.2)	1.23	0.84–1.78	0.28
Atrio-ventricular block	4 (0.8)	6 (0.5)	1.79	0.47–6.90	0.40
Autoimmune disease	8 (1.6)	15 (1.1)	1.90	0.76–4.74	0.17
Cardiomyopathy or heart failure	17 (3.5)	31 (2.3)	1.43	0.76–2.71	0.27
Myocarditis	45 (9.1)	158 (11.9)	0.82	0.57–1.18	0.28
Ischemic stroke or transient ischemic attack	0 (0.0)	1 (0.1)	-	-	1.00
Pericarditis	0 (0.0)	1 (0.1)	-	-	1.00
Pulmonary embolism	0 (0.0)	3 (0.2)	-	-	1.00
Embolism or arterial thrombosis	0 (0.0)	1 (0.1)	-	-	1.00
Other reason	284 (57.7)	770 (58.1)	1.03	0.82–1.28	0.82

CI—confidence interval; OR—odds ratio; * females vs. males adjusted for age.

**Table 4 jcm-10-05502-t004:** Short- and long-term observed mortality of patients aged 20 years or younger hospitalized with myocarditis according to sex and age group.

Follow-Up	Age Group	Mortality (%)		*p*-Value
		Females	Males	
In-hospital	0–5	2.5%	1.0%	0.21
	6–10	2.1%	1.6%	1.00
	11–15	0.6%	0.9%	1.00
	16–20	0.7%	0.2%	0.40
30 days	0–5	2.5%	1.5%	0.50
	6–10	2.1%	3.2%	0.89
	11–15	1.3%	1.2%	1.00
	16–20	1.1%	0.2%	0.08
1 year	0–5	5.1%	1.9%	0.03
	6–10	4.3%	3.8%	1.00
	11–15	1.9%	1.6%	1.00
	16–20	1.1%	0.3%	0.21
3 years	0–5	5.8%	2.3%	0.04
	6–10	1.5%	3.9%	0.62
	11–15	0.9%	1.7%	0.89
	16–20	1.3%	1.1%	1.00
5 years	0–5	6.4%	1.3%	0.01
	6–10	2.4%	4.7%	0.89
	11–15	1.4%	3.9%	0.55
	16–20	1.9%	1.9%	1.00

**Table 5 jcm-10-05502-t005:** Relative survival of patients aged 20 years or younger hospitalized with myocarditis according to sex and age group.

Gender	Outcome	Age Group			
		Relative Survival			
		0–5	6–10	11–15	16–20
Males	30 days	0.99 (0.98–1.00)	0.98 (0.96–0.99)	0.99 (0.98–1.00)	0.998 (0.997–1.00)
	1 year	0.99 (0.98–1.00)	0.97 (0.95–0.996)	0.99 (0.97-0.998)	0.998 (0.995–1.00)
	3 years	0.99 (0.97–1.00)	0.97 (0.94–0.99)	0.98 (0.97–0.997)	0.99 (0.99–0.999)
	5 years	0.99 (0.97–1.00)	0.96 (0.93–0.99)	0.97 (0.94–0.99)	0.99 (0.99–0.998)
Females	30 days	0.98 (0.96–0.99)	0.98 (0.95–1.01)	0.99 (0.97–1.01)	0.99 (0.98–1.00)
	1 year	0.95 (0.93–0.98)	0.96 (0.92–0.998)	0.98 (0.96–1.00)	0.99 (0.98–1.00)
	3 years	0.95 (0.92–0.97)	0.96 (0.92–0.998)	0.98 (0.96–1.00)	0.99 (0.98–1.00)
	5 years	0.95 (0.92–0.97)	0.96 (0.92–0.999)	0.98 (0.96–1.00)	0.99 (0.97–1.00)

## Data Availability

Data are available on request.

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
