# Peer review of "Sex Differences in Incidence, Clinical Characteristics and Outcomes in Children and Young Adults Hospitalized for Clinically Suspected Myocarditis in the Last Ten Years—Data from the MYO-PL Nationwide Database"

_jcm, 2021, doi:10.3390/jcm10235502_

Round 1

Reviewer 1 Report

Introduction: Authors state "there is still a substantial gap in the current knowledge of epidemiology, diagnosis, treatment and true course of the disease not only in adults but also in the paediatric/ young 47
adult population." however, there is abundant literature available regarding incidence of myocarditis from different countries. Please state whether the authors refer to their country. For example Klugman et al and Vasudev et al have published studies on pediatric myocarditis using large databases. There have been reports published from Australia and Finland by Nugent et al and Arola et al respectively. 

Line 49 - authors need to clarify whether this is true  for their region or for every where else. If this is general statement this is not true. 

Line 50-55: please state references. See comment above please state that incidence data is available from other countries (references) however, similar data is lacking regionally. 

line 57 - Again not true or authors have not conducted thorough literature review. Vasudev et al analyzed data on 60 million hospitalization from 2006-2016 using large data representing 95% of the United states. 

Line 65: I came across a paper titled from polad "Occurrence, Trends, Management and Outcomes of Patients Hospitalized with Clinically Suspected Myocarditis—Ten-Year Perspectives from the MYO-PL Nationwide Database". Authors have reported incidence data. 

Methods: Line 80 - "Based on the NHF data, we derived hospitalizations due to myocarditis reported in the 80
years 2009-2020 with the following ICD-10 (the International Classification of Diseases 81
and Related Health Problems, 10th Revision) codes: I40, I40.0, I40.1, I40.8, I40.9, I41, I41.0, 82
I41.1, I41.2, I41.8, I51.4 and B33.2." please state reference. Please elaborate whether this is discharge abstract. So any diagnosis field is searched or only those with first (principal) diagnosis of myocarditis searched. 

Statistical analysis: 
line 107 - Did authors check distribution of data? Please mention how distribution was checked and therefore based on distribution of data mean and median were reported. 
Further, to check for difference, if it is non normal distribution, will reqire to report non parametric test. 

line 112 - how were variables for mortality selected? 

How was incidence analyzed? What was the denominator that was used.

Which test were used for checking trends? 
Chi-square p-value is different when chi-square p-value is generated using ptrend command. 
I would suggest using linear regression. 

I would recommend showing the Kaplan Meier survival curve as figure. This will allow readers if there is overlap. Also how did authors check if survival among male and female was different? 

Discussion Line 223 - Discussion needs to be revamped. 
First sentence is again untrue. 

Line 230 - again this is untrue. Vasudev et al showed myocarditis incidence higher among boys compared to girls. 

Rest of discussion needs to be revamped. Authors need to highlight the reasons for differences in their study vs those that are already published and also come up with reasons. 

- I see that trends for incidence of myocarditis have been declining in all groups except for 16-20 years. 
What is the reason for such finding. Vasudev et al showed that incidence is on rise. This finding is different. 

Additionally, CMR or EMB are gold standard. However, their utilization in the current study population is limited. This will affect the interval validty of study as we are relying on clinical judgement/Echo finding to diagnose. 

Need figure for population derivation. 
Baseline table showing distribution of data. 

Introduction: hypothesis need to be clearly stated. 
Results and Discussion format should flow in order based on the order of findings. 

- incidence among male and female.
- In hospital outcomes. Predictors of mortality. 
- management strategies among pediatric myocarditis 
, etc. 

Reviewer 2 Report

The authors presented a large number of pediatric patients hospitalized due to myocarditis between 2011 and 2019. They showed significant age- and sex-related differences in the incidence, clinical characteristics, and disease short-, mid- and long-term outcomes.

Although, I have some suggestions:

  • Page 2, lines 83 – 85 – if the diagnostic criteria of myocarditis were only clinician-dependent, that means authors might include many patients with no cardiac-related thoracic pain in the study. So, numerous patients might not have acute myocarditis, but some other clinical state with symptoms similar to acute myocarditis. According to ESC recommendations, at least one symptom (chest pain, palpitations, fatigue, dyspnoea, signs of cardiogenic shock) and at least one of the following diagnostic criteria are required to confirm the diagnosis: 1) increased value of cardio specific enzymes, 2) abnormalities in ECG record, 3) impaired left ventricular (LV) kinetics determined by echocardiography and cardiac nuclear magnetic resonance imaging (CMR), and 4) presence of inflammation on CMR findings. In asymptomatic patients, at least two diagnostic criteria are required. Consequently, authors should revise their database, while some patients might not have acute myocarditis.
  • Page 5, line 161 – it is an inappropriate title for this paragraph
  • Page 5, lines 171 – 174 – information about the frequency of performed analyses (CRP, troponin and brain natriuretic peptides) is not significant for future clinical practice. It will be more valuable to present age-related differences between values of those laboratory markers
  • How do authors explain why arrhythmias were more frequent in girls than in boys?
  • Why do girls have more common severe clinical presentation and higher mortality rates than boys?

Round 2

Reviewer 1 Report

I commend work of authors, while most of my comments were addressed please find my minor comments below: 

Materials and methods - page 3, line 113. 
"years 2011-2019." If this is correct then it needs to be corrected in Abstract line 22. "0-20 years hospitalized for myocarditis in the years 2009-2020." 

Results - 3.1 line 136. Please correct sentence. Median age was 17 years [Interquartile range (IQR), 13-19] and XX years in males [IQR, ] and XX years among females [IQR,].

I understand that authors have reported chi-sq p-value but to show trend Chi-square p-value is not useful. 
I would suggest Linear regression for continuous variables and Ch-square p-value using P-trend command and report it as P-Trend. Advise using jonckheere-terpstra test since authors report incidence using Polish age standardize data. Without this reporting chi-square p-value is of no value and cannot comment on trends. 

Page 5, figure 2 - advise analyzing ch-square p-value which helps us to state that there was a difference. Without p-value commenting on figure is not helpful. 
